# The use of OpenAlex to produce meaningful bibliometric global overlay maps of science on the individual, institutional, and national levels

Robin Haunschild[ID][1]*, Lutz Bornmann[2]*

1 Max Planck Institute for Solid State Research, Stuttgart, Germany, 2 Science Policy and Strategy Department, Administrative Headquarters of the Max Planck Society, Munich, Germany

* l.bornmann@fkf.mpg.de, bornmann@gv.mpg.de (LB); r.haunschild@fkf.mpg.de (RH)

**Data Availability Statement:** The data used are freely available from OpenAlex, see: https://openalex.org/. The produced base maps are available at: http://ivs.fkf.mpg.de/global_maps_

## Abstract

The Social Systems Citations Theory (SSCT) is the most recent theory of citations integrating previous theories. It focuses on communications in science that are formally manifested as publications and citations in scientific communication networks. These networks can be observed and empirically studied by using science maps. Science maps typically visualize networks of communication elements such as key words, cited references, and subject areas. In this study, a procedure to create global overlay maps using OpenAlex is proposed. It is an important advantage of OpenAlex publication and citation data that they are freely available. Overlay maps visualize how the overlaid data (e.g., research of an institution) are positioned in the whole science system (the base map). Six different base maps are provided to the user for their own applications. Using one of these base maps, example overlay maps for two individuals (the authors of this paper) and four research institutions are shown and discussed. A method for normalizing the overlay data is also proposed that can be used for the comparison of two different overlaid units. Overlay maps using raw overlay data display general concepts more pronounced than specific concepts. It is the other way around with their counterparts using normalized overlay data. Advantages and limitations of the proposed overlay approach based on OpenAlex are discussed.

## 1. Introduction

Since the establishment of networks is characteristic for a modern science system [1]–"science has become more interconnected over time" [2]–bibliometrics results have been more frequently presented as science maps over the years. The annual releases of the Leiden ranking, https://www.leidenranking.com, [3] and the institutional excellence maps, https://www.excellencemapping.net, [4] are good examples for this trend: institutional performance data are no longer exclusively presented in tables and lists, but also visualized as science maps. According to Bales et al. [5], science maps are "spatial representations of how disciplines, fields, specialties, and individual documents or authors are related to one another". Moed [6] defines science mapping as "the development and application of computational techniques for

OpenAlex/ (DOI: 10.17617/1.daf7-fq06). The base maps can be downloaded as a zip file: http://ivs.fkf.mpg.de/global_maps_OpenAlex/global_base_maps_levels_0_1_2_OpenAlex_Aug23.zip".

**Funding:** The author(s) received no specific funding for this work.

**Competing interests:** "Both authors currently serve as academic editors for PLOS ONE. This does not alter our adherence to PLOS ONE policies on sharing data and materials."

the visualization, analysis, and modeling of a broad range of scientific and technological activities as a whole" (p. 76). In many science maps, community-findings algorithms are used to identify and mark (color) cliques on the author, institutional or national level [7]. Several programs exist today that can be used to map science and to apply the community-findings algorithms [5].

The popularity of science maps in empirical bibliometrics today has also settled into theorizing on bibliometrics. Science maps conceptualize science as a communication system, where "information obtained at certain nodes is transmitted to other nodes" (1, p. 113). Citations play an important role in these communication systems, since they form the bridges between published research [8] and may point out research communities and their connections. Tahamtan and Bornmann [9] recently introduced a new citation theory that is based on the social systems theory proposed by Niklas Luhmann [10]. The Social Systems Citations Theory (SSCT) focuses on communications in science that are formally manifested as publications and citations in scientific communication networks. These networks can be observed and empirically studied by using science maps. The SSCT integrates previous citation theories (especially the normative and social constructivist approaches) and avoids their specific biases and limitations (e.g., each theory only focuses on a limited set of motives to cite publications, although other motives exist). The SSCT regards citations as elements in (formal) science communication networks that can be analyzed independently from the citing authors (9). Citing authors are not part of the social systems (that constitute, e.g., science maps), but are–as psychic (mental) systems–elements in their environment.

In the SSCT, citation networks are constituted as autopoietic systems. The autopoietic systems are operatively closed and (re-)produce publications and citations on their own terms: Every new publication includes citations and open questions for subsequent research and publications [9]. Luhmann [10] positions psychic systems (i.e., authors) outside social systems: Psychic systems (i.e., authors) are not part, but stimulate the citation links (formal communications) in science networks. The stimulated citation links constitute the matrix of cited and citing publications in the social system. This matrix can be denoted as the web of knowledge: Karl R. Popper's world 3 [11]. The SSCT displaces humans' motives and reasons to cite certain publications in the periphery of the social system (i.e., in the psychic systems). The psychic and social systems operate autonomously but also interact with each other: The mental systems of authors are structurally necessary for establishing citation links between publications in the science system; both systems are structurally coupled [9].

Science maps are conceptually rooted in the SSCT [9, 12]. Although science cannot be imagined without creative scientists (i.e., mental systems), science maps represent (visualize) processes on the communication level (i.e., social systems). This study deals with a specific form of science maps: global science overlay maps. These maps apply base maps (usually including the whole science system), on which certain overlay data are presented (e.g., the publications of an institution). Overlay maps visualize how the overlaid data (e.g., research of an institution) are positioned in the whole science system. For example, field-specific activities of institutions can be made visible using global base and overlay maps. We introduce in this study the use of OpenAlex [13] data (a new bibliometric database that is freely available) to build meaningful global science overlay maps that can be overlaid by the user with own data. For example, specific institutional overlay data can be used in combination with the base maps from this study to explore the field-specific activities of the institution.

## 2. Global science overlay maps

The visualization of data as overlay maps is a development in bibliometrics that started around a decade ago [14]. The generation of global overlay maps include two steps: In the first step,

meaningful global base maps are developed that can be used as a basis for overlays with specific data (e.g., data from an institution or country). Meaningful global base maps position certain elements (e.g., publications or fields) in such a way that most of the elements are well visible to an observer. In the second step, the data of specific interest are downloaded from databases (e.g., Web of Science, WoS, Clarivate, or Scopus, Elsevier) and overlaid on the global base map. Overlay maps have been widely used in different contexts. For example, some authors have applied the technique to overlay bibliometric data on Google maps [15–17].

The global overlay mapping technique was initially introduced by Boyack [18]. Klavans and Boyack [19] were the first to propose a global map of science that can be used as base map for being overlaid. Klavans and Boyack [20] provide "a deductive argument that global mapping should create more accurate partitions of a research field than does local mapping" (p. 1). The global overlay mapping technique was further developed, e.g., by Rafols, Porter, and Leydesdorff [21], Leydesdorff and Rafols [22], and Leydesdorff, Rafols, and Chen [14]. The authors proposed to produce interactive overlays that can be presented online. Most of the global base maps and overlay maps have been generated based on WoS data.

Leydesdorff, de Moya-Anegón, and Guerrero-Bote [23] introduced global base maps that can be overlaid with Scopus data. Mono-disciplinary databases have also been used to produce global overlay maps: Global base maps have been published on the basis of the Medical Subject Headings (MeSH) used in the PubMed database [24]. Bornmann and Haunschild [25] transferred the overlay technique from bibliometrics to the area of alternative metrics (altmetrics). They used Mendeley data (Mendeley is an online reference manager) to produce global science maps that reflect reading of papers (instead of citing of papers). Sjögårde [26] proposed improvements of global overlay maps including both (i) broad disciplines for an overview of science and (ii) granular levels for more detailed information. In 2023, Kevin Boyack and Richard Klavans received the Derek John de Solla Price Medal for their outstanding contribution to the field of scientometrics [27]. Kevin Boyack emphasized the importance of his research on global overlay maps in his speech during the awards ceremony.

The popular VOSviewer software [28] can be used to produce global overlay maps: "VOSviewer . . . supports overlay visualizations. In an overlay visualization, the color of a node indicates a certain property of the node. For instance, nodes may represent journals and the color of a node may indicate the number of times a journal has been cited" [29]. One needs to construct the network of the relevant entities with their association strengths and construct the global base map using VOSviewer. Having large amounts of publication data, the construction of a global base map is always time consuming and resource demanding in contrast to the overlay procedure. We are not aware of any openly and freely accessible global base map that is based on a multidisciplinary bibliometric database, such as WoS or Scopus that can be used to overlay own data. Thus, in this contribution, we aim to provide openly and freely accessible global base maps using all publications indexed in OpenAlex that can be used by anyone to overlay particular data of interest (e.g., publication sets of single institutions or researchers). OpenAlex is a free and open catalog of the global research system (see https://openalex.org/).

## 3. Methods

### 3.1 Datasets

Currently, OpenAlex (14) provides snapshots once a month. We used the snapshot from August 2023 for our analyses as available from the German Kompetenznetzwerk Bibliometrie (see https://bibliometrie.info/). The snapshot contains 243,053,925 documents. We used the following different time periods for our analyses: (i) 1800–2022 with 237,876,541 documents, (ii) 2008–2022 with 134,092,007, (iii) 2013–2022 with 95,438,459 documents, (iv) 2018–2022

with 47,665,990 documents, and (v) 2022 with 8,496,167. We constructed global maps of science using these time periods. Depending on the publication years of the overlay maps, the user can select one of these base maps. The OpenAlex snapshot we used provides concepts on six different levels (from 0 to 5). As of February 2024, OpenAlex also comprises four domains, 26 fields, 252 subfields, and 4516 topics (see: https://help.openalex.org/how-it-works/topics). The domains, fields, and subfields are too broad for the granularity that we want for our base maps. The topics do not seem to be mature enough, for example different topics with apparently the same content exist: "Coronavirus Disease 2019 Research" and "Coronavirus Disease 2019" as well as "Lithium-ion Battery Technology" and "Lithium Battery Technologies". Furthermore, many topics have very long names so that only few topic names could be displayed on a map. Thus, we used concepts algorithmically assigned by OpenAlex to documents as nodes on the map.

Base maps are most appropriate if they enable insights in themes, topics or areas of worldwide research or research of certain units (such as individual countries or institutions). All publications in OpenAlex are assigned to concepts that can be used for this purpose. Concepts are defined as follows: "Concepts are abstract ideas that works are about. OpenAlex indexes about 65k concepts. The CEID [Canonical External ID] for OpenAlex concepts is the Wikidata ID, and concepts have one, because all OpenAlex concepts are also Wikidata concepts." [13]. OpenAlex has built upon the decommissioned Microsoft Academic Graph (MAG). Scheidsteger and Haunschild [30] have investigated the differences and commonalities in concept assignments between MAG and OpenAlex. They reported that OpenAlex has a higher percentage of documents with at least one concept than MAG. A document can be "tagged with multiple concepts, based on its title and abstract. The tagging is done using an automated classifier that was trained on MAG's corpus" [13].

We explored different possibilities of using each level separately or combining concepts of different levels to produce well suited base maps. The maps based on concepts of the top three levels (i.e., 0, 1, and 2) produced the best maps to have good insights into worldwide research. Thus, we only include documents that have a concept of level 0, 1, or 2 assigned to them. There are 19 concepts on level 0, 284 concepts on level 1, and 21,460 concepts on level 2. That are 237,830,057 documents for the time period 1800–2022 assigned to at least one of 21,758 concepts; 134,054,634 documents for time period 2008–2022 assigned to at least one of 21,758 concepts; 95,406,638 documents for time period 2013–2022 assigned to at least one of 21,758 concepts; 47,641,330 documents for time period 2018–2022 assigned to at least one of 21,756 concepts; 8,478,584 documents for the publication year 2022 assigned to at least one of 21,715 concepts. As we want to provide global maps, we do not impose any restrictions on document types.

## 3.2 Statistics

For calculating the positions of the concepts on the base maps, we used direct citation relations between the documents that belong to these concepts. For all maps but one, we used a citation window of five years plus the publication year. We produced one version of the map that includes the time period 1800–2022 with a citation window of 30 years plus the publication year. We calculated the citation relations as reference relations so that the citation window can be equally long for each document, i.e., citation relations between 1992 (2017) and 2022 were considered for documents published in 2022, and citation relations between 1770 (1795) and 1800 were considered for documents published in 1800 for a citation window of 30 (5) years. As an example, we provide the SQL for the time period 2008–2022:

```
select cd.id cited_concept, cg.id citing_concept, count(*) cit_rels
from fiz_openalex_rep_20230819_openbib.works_referenced_works r join
fiz_openalex_rep_20230819_openbib.works_concepts wcd on referenced_
work_id=wcd.work_id join fiz_openalex_rep_20230819_openbib.concepts
cd on cd.id=wcd.concept_id join fiz_openalex_rep_20230819_openbib.
works wd on wd.id=wcd.work_id join fiz_openalex_rep_20230819_openbib.
works_concepts wcg on r.work_id=wcg.work_id join fiz_openalex_rep_
20230819_openbib.concepts cg on cg.id=wcg.concept_id join fiz_
openalex_rep_20230819_openbib.works wg on wg.id=wcg.work_id where cd.
level between 0 and 2 and cg.level between 0 and 2 and wg.publication_
year-wd.publication_year between 0 and 5 and wg.publication_year
between 2008 and 2022 group by cd.id, cg.id order by cd.id, cg.id;
```

When running this SQL in another environment, the schema name (printed in bold) has to be adjusted. Only the publication years and publication year differences need to be adjusted when the other base maps proposed in this study should be created. The SQL exports the citation relations between the concepts from an OpenAlex snapshot in a PostgreSQL database. This procedure needs large amounts of hard disc storage and large amounts of memory to store and analyze the OpenAlex data especially using longer time periods. The SQLs were run on the KB server with 384 GB RAM, 46 TB hard disc, and an Intel(R) Gold-6354 CPU @ 3.0GHz. We expect that most of the hard disc space is necessary in the KB for other data (e.g., WoS and Scopus). Thus, less hard disc space should be needed to complete the SQL. The most demanding SQL (1800–2022 with a thirty-year citation window) needed about six hours. The tables resulting from the abovementioned SQL and adjusted versions for the other maps were exported from the database and imported into VOSviewer (via "Create", "Create a map based on network data", "VOSviewer network file"). Map files were saved within VOSviewer. Analysis of the citation relations with VOSviewer needs large amounts of memory. We used our calculation server at the Max Planck Institute for Solid State Research (with 512 GB RAM and an Intel(R) Xeon(R) E5-2640 v3 CPU @ 2.60GHz) with 200 GB RAM requested in the VOSviewer execution for creating the base maps from the SQL export. The most demanding base map (1800–2022 with a thirty-year citation window) needed less than one hour. Thus far, the global base maps used the concept IDs as node names. We replaced the ID labels with the corresponding concept labels (i.e., names) to produce interpretable maps. These map files were opened in VOSviewer to produce the images of the base maps. We used the following parameters to construct the base maps: clustering resolution of 1.25, minimum cluster size of 500, label size variation of 0.2, and scale of 0.5.

The global base maps were overlaid with different datasets in this study to demonstrate the overlay technique using OpenAlex data. We produced two overlay maps for researchers and four maps for institutions. We chose six examples that are well interpretable for us: (i) documents assigned to the OpenAlex author ID of the first author of this paper (RH), (ii) documents assigned to the OpenAlex author ID of the second author of this paper (LB), (iii) documents assigned to the OpenAlex institution ID of the Max Planck Institute for Solid State Research (MPI-FKF), (iv) the Max Planck Institute for Plasma Physics (MPI-IPP), (v) the Max Planck Institute for Psycholinguistics (MPI-PL), and (vi) the Max Planck Institute for Informatics (MPI-I). We present and discuss the example maps with respect to advantages and disadvantages of the overlay technique using OpenAlex data. We kept the VOSviewer parameters label size variation of 0.2 and scale of 0.5 from the base maps for the overlay maps of RH and LB. However, we used size variation of 0.4 and scale of 1.0 for the maps of MPI-FKF, MPI-IPP, MPI-PL, and MPI-I to have a better distinction between the node sizes of the institutional maps. We recommend that the optimal values for size variation and scale are chosen for the

overlay maps by the users. However, if two overlay maps are compared, the maps should be created using the same parameters.

Data to overlay on the global base maps can be exported from the OpenAlex web-interface, downloaded from the OpenAlex API, or exported from an OpenAlex snapshot. Readers can download the global base maps and the overlay maps discussed in this paper at: https://doi.org/10.17617/1.daf7-fq06. Thus, readers can experiment with other VOSviewer parameters using our overlay maps. For overlaying data on the global base maps, the following steps should be done: (1) The column "weight<papers>" needs to be replaced with the data to be overlaid. (2) The labels of the concepts without documents in the focus dataset should be removed, and a color that does not occur in the global base maps should be chosen. (3) The value in the column "weight<papers>" for concepts without papers from the focal unit should be removed.

For the comparison of overlay maps, it is necessary to apply normalization procedures to the overlay data. We propose to normalize the overlay data by dividing the proportion of documents per concept in the focus dataset by the proportion of documents per concept in OpenAlex:

$$N_{Wl} = \sum_c N_{cWl}$$

$$N_{Ul} = \sum_c N_{cUl}$$

$$p_{cWl} = {N_{cWl}}/{N_{Wl}}$$

$$p_{cUl} = {N_{cUl}}/{N_{Ul}}$$

$$a_{cUl} = {p_{cUl}}/{p_{cWl}}$$

Here, $N_{Wl}$ and $N_{Ul}$ are the total number of documents of the world ($W$) and the focal unit ($U$) on the same level $l$ as the concept $c$. The number of documents assigned to specific concepts ($c$) of the world and the focal unit are $N_{cWl}$ and $N_{cUl}$. The proportions of documents assigned to specific concepts of the world and the focal unit are $p_{cWl}$ and $p_{cUl}$. The activity, i.e., normalized proportion of documents assigned to specific concepts of the focal unit is $a_{cUl}$. These activities replace the values in the column "weight<papers>". This normalization procedure produces the ratio of the proportion of documents assigned to a certain concept of the focal unit to the world's proportion of papers assigned to the same concept. This means that equally sized nodes on the raw maps will be of different sizes on the normalized maps if the world's proportions for these concepts are different.

Global reference data needed for the normalization procedure can be retrieved from the OpenAlex API via the following three API calls:

- https://api.openalex.org/concepts?filter=level:0

- https://api.openalex.org/concepts?filter=level:1

- https://api.openalex.org/concepts?filter=level:2

For obtaining overlay data for authors or institutions, one needs the author or institution ID from OpenAlex and can use the corresponding API call:

- Authors API call example: https://api.openalex.org/works?filter=author.id:A5004115158

- Institutions API call example: https://api.openalex.org/works?filter=institutions.id: I4210088365

The API results need to be gathered by concept ID. Aggregation of the number of papers per concept provides the corresponding values $N_{Wl}$ and $N_{Ul}$ that either can be used as overlay data or for the calculation of the normalized variant ($a_{cUl}$):

- Authors API call example: https://api.openalex.org/works?filter=author.id: A5004115158&group_by=concepts.id

- Institutions API call example: https://api.openalex.org/works?filter=institutions.id: I4210088365&group_by=concepts.id

## 4. Results

In section 4.1, the base maps are explained that can be used for overlay maps. The examples in section 4.2 demonstrate how certain overlays can be applied to the base maps and how the maps can be interpreted.

### 4.1 Base maps

Fig 1 shows six global base maps that we produced in this study. The clusters were determined by the VOSviewer algorithm association strength using citation relations. This gave rise to six different clusters. The cluster colors were assigned by the cluster size: The largest cluster is colored in orange, the second largest in green, the third largest in blue, the fourth largest in yellow, the fifth largest in pink, and the smallest cluster in light blue. The cluster size is determined by the number of concepts that belong to the cluster. As the maps in Fig 1 demonstrate, all maps reveal rather similar patterns. The most visible difference is the interchange of the cluster color assignments of yellow and blue. The clusters are of rather similar size, e.g., the blue and yellow cluster contain 3,232 and 3,144 concepts in panel A and 3,299 and 3,111 concepts in panel B. Depending on the size rank of the clusters in the maps, either the one cluster is yellow (blue) or blue (yellow).

Larger map differences can be observed in the case of the global base maps of the two shortest time periods (2018–2022 and 2022 in panels E and F). More concepts are located differently compared to the other maps, e.g., the concepts Voltage and Particle physics are located much closer together in panel F than in panel C. However, these differences are only visible when zooming into the maps using VOSviewer. Since a few concepts are located further away from most concepts, e.g., Mucoproteins and Videotex, panels E and F show zoomed images of the maps. The scattering can be explained by too few direct citation relations within the shorter time periods which lead to very low association strengths.

In the following, we focus on the global base map within the time period 2008–2022. It is shown larger in Fig 2 with the main scientific fields annotated. The orange cluster contains 6,788 concepts mainly from the social sciences and the humanities. The green cluster mainly contains 4,210 medicinal concepts. The blue cluster contains 3,247 concepts from the fields of physics and engineering. The yellow cluster contains 3,153 concepts from the fields of mathematics, computer sciences, and theoretical physics. The pink cluster contains 2,825 concepts mainly related to biology. The light blue cluster covers 1,534 concepts from chemistry and material sciences.

### 4.2 Examples for overlay maps

In this section, we show some overlay maps as examples by using the base maps presented in the previous section. On the one hand, the example maps are intended to demonstrate the

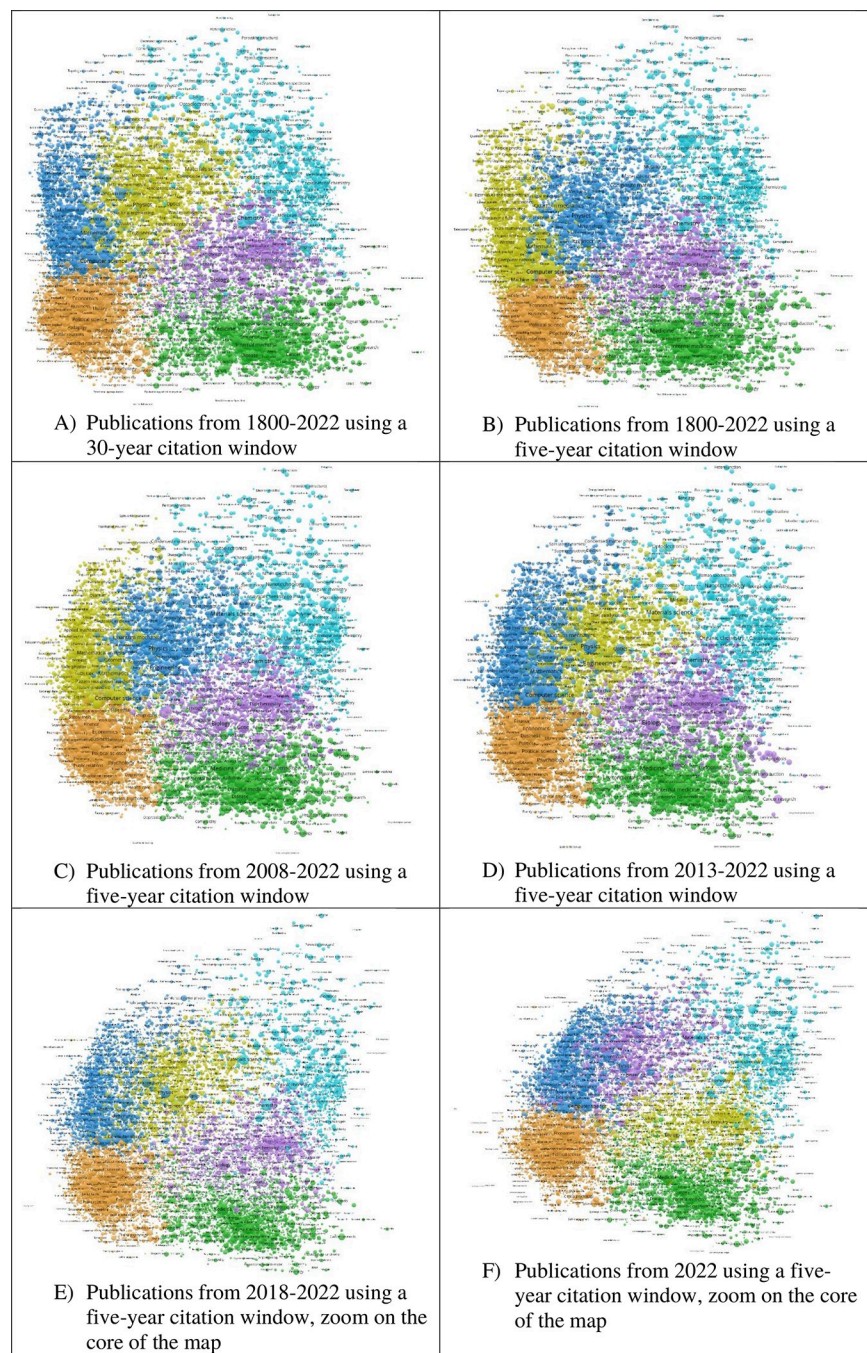

**Fig 1.** Six different global base maps using different time periods (B-F) and using a longer citation window for the longest time period (A).

usefulness of the overlay approach. On the other hand, we discuss based on the specific overlay maps whether the overlay approach using OpenAlex data produces reliable and meaningful results. We generated overlay maps that can be well interpreted by at least one of us. The approach developed in this study stands or falls with the quality of the concepts provided by OpenAlex, since the most important information on the overlaid research are the labels from the concepts.

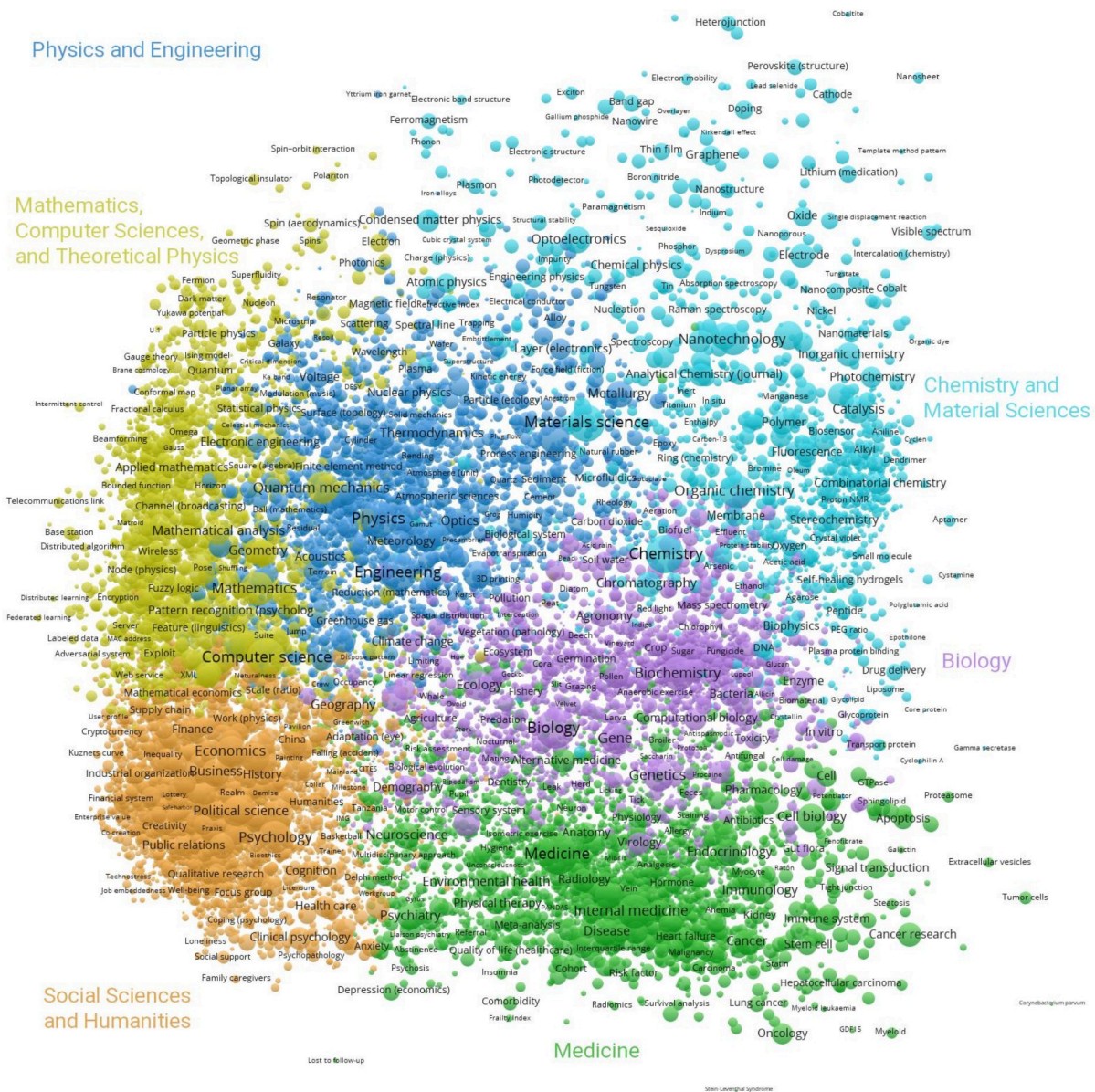

**Fig 2. Global base map using the time period 2008–2022.**

Fig 3 shows the overlay maps for six different focus datasets: (i) Panel A shows the global overlay map for documents assigned to the OpenAlex author ID of the first author of this paper (RH). (ii) Panel B shows the global overlay map for documents assigned to the OpenAlex author ID of the second author of this paper (LB). (iii) Panel C shows the global overlay map for documents assigned to the OpenAlex institution ID of the Max Planck Institute for Solid State Research (MPI-FKF). (iv) Panel D shows the global overlay map for documents assigned to the OpenAlex institution ID of the Max Planck Institute for Plasma Physics (MPI-IPP). (v) Panel E shows the global overlay map for documents assigned to the OpenAlex institution ID of the Max Planck Institute for Psycholinguistics (MPI-PL). (vi) Panel F shows the global overlay map for documents assigned to the OpenAlex institution ID of the Max Planck Institute for Informatics (MPI-I).

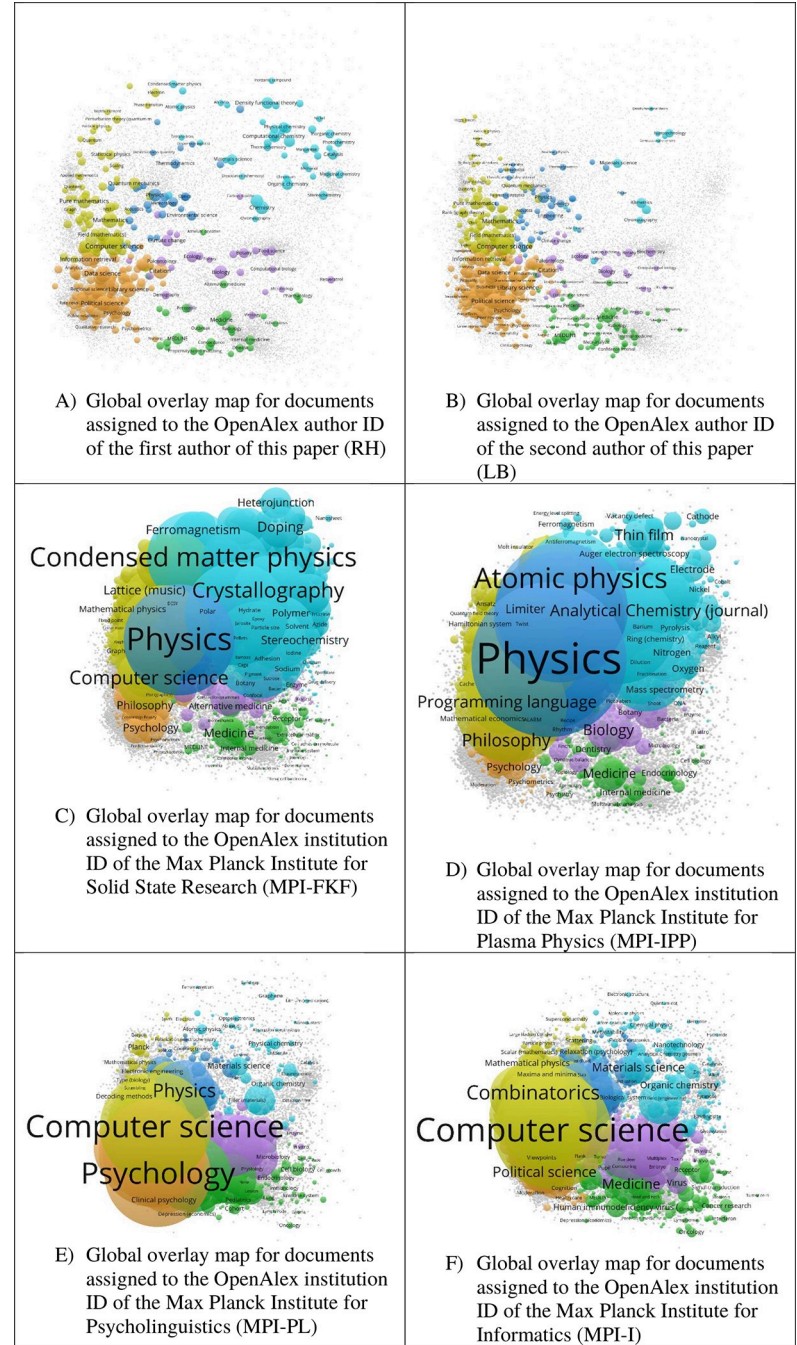

**Fig 3. Six global overlay maps using different focus datasets.**

Before conducting research in bibliometrics, RH (see https://www.researchgate.net/profile/Robin_Haunschild) was occupied with research in computational and theoretical chemistry (see Fig 3A). This is reflected in the pronounced light blue nodes, e.g., Computational chemistry, Density functional theory, and Catalysis, and some of the yellow nodes, e.g., Perturbation theory (quantum mechanics). However, we discovered some weaknesses of the labeling of the concepts by OpenAlex, e.g., the concept Spin (aerodynamics), the unlabeled node left of

Electron, has a wrong explaining term in parentheses. At least in this case, the documents are related to spins of electrons. The location of this concept in close vicinity to other physical concepts, e.g., Electron, Phase transition, and Condensed matter physics, indicates that the general meaning is the one of the spin of physical particles, e.g., electrons. Actually, the concept Spins is also in close vicinity of the concept Spin (aerodynamics). Probably, these two concepts should be merged in OpenAlex to the Concept Spin. If wished so, one can do this oneself by adding the number of documents of the two concepts and modifying the label of the concept. Preferably, such inaccuracies should be eliminated by the providers of OpenAlex in the database.

RH's current research activities can be seen in the pronounced orange cluster, e.g., Data science, Citation, Library science, and Information retrieval. However, some bibliometric papers seem to also be assigned to concepts such as Computer science, Mathematics, and Field (mathematics). The latter most probably is related to field normalization in bibliometrics. Without insight into the dataset, the explanatory addition "mathematics" in parentheses would be more confusing than helpful. That many documents are assigned to Computer Science can partly be explained by the classification of most papers published in the journal *Scientometrics*. 88.9% of the *Scientometrics* papers are assigned to the concept Computer science. About half of the papers published in the journal *Scientometrics* are assigned to the concepts Political science and Library science. This also explains the surprisingly pronounced node Political science in Fig 3A. The green and pink nodes are partly due to bibliometric papers about environmental, medical, and biological fields, e.g., Climate change, Biology, and Medicine. This also holds true for some of the blue nodes, e.g., Meteorology.

Panel B of Fig 3 shows the global overlay map for documents assigned to the OpenAlex author ID of the second author of this paper (LB, see https://www.researchgate.net/profile/Lutz-Bornmann). The difference to the map of RH (Panel A) is clearly visible: whereas RH has been active in two areas in his career (chemistry/physics and social sciences), LB has been active only in the social sciences. Panel B reflects broader topics or areas in which LB is active in research such as library science, psychology, and data science. The concept Peer review is visible on the map but could be shown more prominently, since LB has published many papers on peer review processes. Although some papers published by LB can be classified as political science (as outlined on the map), political science is emphasized too prominently. The same is true with Mathematics, Computer science, and Information retrieval. Similar to the map of RH, some information which are in parenthesis in the concept labels do not reflect LB's research such as Big Bang (financial markets) and Rank (graph theory).

Since the MPI-FKF (see https://www.fkf.mpg.de/en) is broadly covering chemistry and physics from a theoretical and an experimental perspective, the main nodes in panel C of Fig 3 appear in the yellow (mathematics, computer sciences, and theoretical physics), blue (physics and engineering), and light blue (chemistry and material sciences) clusters. The largest concepts, e.g., Physics, Condensed matter physics, and Crystallography, clearly represent some of the main research activities of the MPI-FKF. Like in the maps for RH and LB, we also see in the MPI-FKF map concepts with confusing additions in parentheses, e.g., Lattice (Music). Most of the MPI-FKF papers assigned to this concept are about crystal structure research by retired directors (e.g., Manuel Cardona, Arndt Simon, and Martin Jansen).

The nodes in the other clusters are somewhat surprising. Some of the nodes in the orange, green, and pink clusters might be due to the research activities of the Information Retrieval Service (IVS-CPT). The group provides scientometrics service (and performs research in this field) for the institutes belonging to the Chemistry, Physics, and Technology Section of the Max Planck Society. The IVS-CPT is part of the MPI-FKF, and RH is head of the IVS-CPT. For example, the typical nodes for bibliometrics, such as MEDLINE and Citation, appear not

only on panel C, but also on the overlay map of RH. Some nodes, e.g., In vivo and Immune system, do not appear on the overlay map of RH's documents. Such nodes probably indicate a problematic assignment of concepts to documents in OpenAlex. The rather prominent concept Medicine is mainly due to wrong assignments of papers in OpenAlex. The most frequent author of the MPI-FKF papers that belong to the concept Medicine is Joachim Maier who has been conducting research in electrochemistry. This is also reflected in the bulk of his MPI-FKF papers that are assigned to the concepts Chemistry, Material Science, Physics, Physical chemistry, Organic chemistry, and Electrode.

The MPI-IPP (see https://www.ipp.mpg.de/en) mainly performs research in plasma physics. Most main nodes in panel D of Fig 3 clearly show the connection to research in plasma physics, e.g., Atomic physics. Plasma (large node unlabeled between Physics and Atomic physics), Material science (large unlabeled node right of Physics), and Engineering (large unlabeled node below Physics). Computer science (e.g., Computer language) is also relevant for plasma physics research: the MPI-IPP's departments Tokamak Theory (see https://www.ipp.mpg.de/ippcms/eng/for/bereiche/tokamak), Stellarator Theory (see https://www.ipp.mpg.de/ippcms/eng/for/bereiche/stellarator), and Numerical Methods in Plasma Physics (see https://www.ipp.mpg.de/ippcms/eng/for/bereiche/numerik) employ computer science in their research activities. We also see specific atoms as concepts, e.g., Barium, Nitrogen, and Oxygen. Those atoms have particular relevance in the MPI-IPP's research in plasma physics.

The MPI-PL's (see https://www.mpi.nl/) research activities in psycholinguistics focus on unravelling the mysteries of language. The research of the institute is located between the social sciences, humanities, and life sciences. This is also reflected by the main nodes in panel E of Fig 3 which are Computer science, Philosophy (large unlabeled orange node between Computer Science and Psychology), Psychology, Neuroscience (large unlabeled green node right of Psychology), Medicine (large green node left of Nose), and Biology (large unlabeled pink node left of Microbiology). Inspecting the map in more detail, we see nodes such as Speech recognition (between Computer Science and Psychology), Sentence, Cognitive science, Comprehension, Literacy, Cognition, and Language model. All of them are too close to Psychology to be labeled using this VOSviewer display parameter set. The nodes have a close connection to the institute's research.

The MPI-I's (see https://www.mpi-inf.mpg.de) research is not only focused on formal computer science but also applications thereof. For example, the MPI-I hosts a research group with the name Computational Biology. This is also reflected in the main nodes of panel F of Fig 3, such as Computer Science, Mathematical Physics, Combinatorics, Medicine, and Biology (large unlabeled pink node above Medicine). We assume that the nodes in the light-blue cluster are related to the activities of the MPI-I in medicinal chemistry. Some of the blue nodes seem to be related to the MPI-I's research regarding computer science related to biomechanics.

We presented six overlay maps as examples by using the base maps presented in the previous section. Most concepts shown on the overlay maps can be well associated with the unit's research areas and thus provide helpful guidance in interpretation of the units' research. This is, however, not always the case. The inspection of the maps revealed that biological and medicinal concepts are emphasized too much on the maps of RH, LB, MPI-FKF, and MPI-PL. RH has checked the 34 documents assigned to his author ID and the concept Biology in OpenAlex (see: https://openalex.org/works?sort=cited_by_count%3Adesc&column=display_name,publication_year,type,open_access.is_oa,cited_by_count&page=1&filter=authorships.author.id%3AA5004115158,concepts.id%3AC86803240). Although the documents were correctly assigned to RH, the assignments of most documents to Biology are at best questionable. Two examples for plain wrong assignments of biological concepts are Haunschild, Janesko, and

Scuseria [31] and Haunschild and Scuseria [32]. The authors have proposed new density functional approximations. Some other assignments besides Biology are also wrong, e.g., Finance, Economics, and Botany. These two documents also bear concepts with questionable explanations in parentheses, e.g., Perturbation (astronomy) and Range (aeronautics). Many other assignments of these two documents are correct yet, e.g., Density functional theory, Hybrid functional, Quantum mechanics, Computational chemistry, and Physics. Another example for problematic concept assignments of OpenAlex to papers can be seen in the MPI-FKF map. The most frequent author of the MPI-FKF papers labelled Alternative medicine is Klaus Kern. The corresponding papers are studies about nanoscale materials (see: https://openalex.org/works?page=1&filter=authorships.institutions.lineage%3AI4210088365,concepts.id%3AC204787440,authorships.author.id%3AA5037008818&sort=cited_by_count%3Adesc&group_by=publication_year,open_access.is_oa,authorships.institutions.lineage,type,concepts.id,authorships.author.id).

Fig 4 shows the normalized versions of the global overlay maps from Fig 3. It seems that the normalized versions of the maps provide more meaningful insight into the units' research than the versions without normalization. Comparing Figs 3 and 4, we see in general that some concepts are more pronounced in the normalized overlay version in Fig 4, and others are less pronounced in the raw overlay version in Fig 3. For example in panels A and B, the concepts Scientometrics, Bibliometrics, and Topic model are much more pronounced in Fig 4 than in Fig 3. The light-blue nodes are much smaller in panel B of Fig 4 than in panel B of Fig 3 in agreement with LB's research focus. Similar observations can be made for the other panels, too. For example, the concepts MEDLINE, Medicine, and Internal medicine are much smaller in panel C of Fig 4 than in panel C of Fig 3. General concepts are more pronounced on the raw maps whereas more specific concepts are more pronounced on the normalized maps.

For example in panel D of Fig 3, the general concepts Physics, Atomic Physics, Philosophy, Programming language, and Biology are much more pronounced than in Fig 4. Instead of the more general concepts, panel D of Fig 4 shows more specific concepts such as Pedestal, Tearing, Impurity, Molybdenum, and Ethyl iodide more pronounced than in panel D of Fig 3. Similarly, panel E of Fig 3 shows Computer science, Psychology, Clinical psychology, and Physics as the most pronounced nodes, whereas panel E of Fig 4 exhibits Language evolution, Mentalization, Cardinal direction, Speech delay, and ABX test as most pronounced labeled nodes. This general difference between the maps in Figs 3 and 4 can also be observed when comparing panel F of both figures: In the case of Fig 3F, the most pronounced labeled concepts are Computer science, Combinatorics, Political Science, Medicine, and Material science, whereas the most pronounced concepts in the case of Fig 4F are Deterministic algorithm, First-order logic, Text retrieval, Body shape, and Turnstile. These changes due to the normalization make sense considering the research areas of the different focal units.

In the normalized versions of the maps, more specific concepts become more pronounced than the more general concepts due to normalization of the overlay data. For example, Panel B of Fig 4 correctly identifies scientometrics and bibliometrics as LB's main areas of research: LB is active in these areas since 2004. Related areas of LB's research such as the investigation of peer review processes with respect to external and predictive validity are also prominently positioned on the map. Many nodes on the normalized map are connected to only one or a few papers: For example, the Hawthorne effect is introduced in Bornmann [33]; hot and cold spots in the US research are the topic of Bornmann and de Moya Anegon [34]; mimicry in science is covered in Bornmann [35]; Leydesdorff, Bornmann and Opthof [36] deal with the $h_\alpha$ index that may consider–as the authors propose–the scientist as chimpanzee or bonobo. Some node labels such as Matrix (chemical analysis) cannot be connected to LB's research (even not by the author himself). Matrix in chemical analysis refers to the environment around an

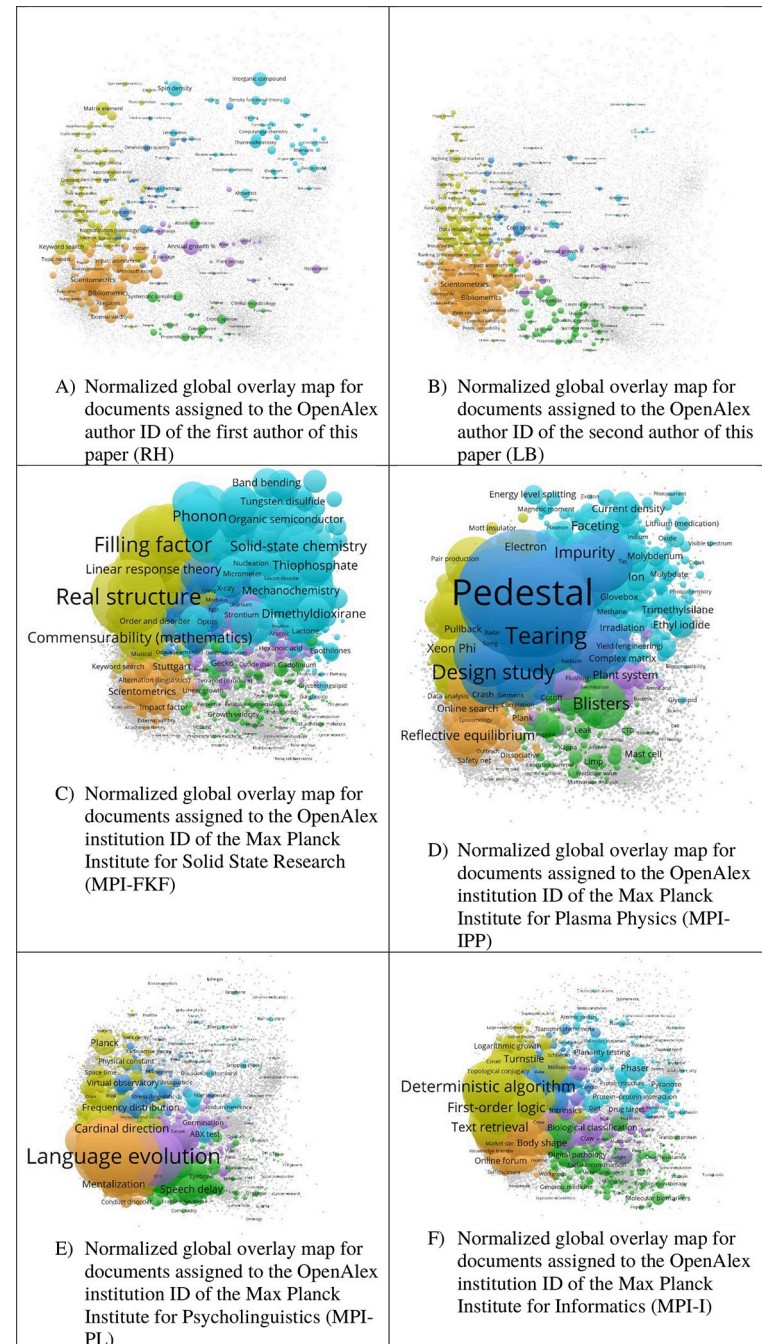

**Fig 4. Six different normalized global overlay maps using different focus datasets.**

analyte (molecule to be identified or analyzed) while a matrix of numbers, e.g., citing and cited papers, seems to be the reason for this label on some of LB's papers.

## 5. Discussion

Although various science maps have been proposed since the beginning of scientometrics research, the visualization of complex bibliometric networks is a more recent trend. One of the

main reasons might be that more and more (big) datasets have been made available and hard- and software for analyzing the data have become more powerful. For van Eck and Waltman [29], the visualization of bibliometric networks "has turned out to be a powerful approach to analyze a large variety of bibliometric networks, ranging from networks of citation relations between publications or journals to networks of co-authorship relations between researchers or networks of co-occurrence relations between keywords . . . At the same time, professional users of bibliometrics, for instance research institutions, funding agencies, and publishers, have become more and more interested in bibliometric network visualizations" (p. 285). Skov, Wang and Andersen [37] discuss how network visualizations can be used for strategic thinking (in research evaluation) by navigating in dynamic research landscapes.

Science maps are conceptually rooted in the SSCT introduced by Tahamtan and Bornmann [9]. Science maps typically visualize networks of communication elements such as key words, cited references, and subject areas. For Chen and Song [38], for example, "science mapping approaches typically aim to represent patterns and trends of the development of science at macroscopic levels such as disciplines and fields over a long period of time" (p. 106). The social systems theory explains society and functional sections of society such as science as social systems that are constituted by communications. The science system and social systems in science constitute a network consisting of interlinked communications that can be made visible by bibliometric networks (science maps). In the research area of science visualization in bibliometrics, the SSCT can be especially used to explain their basic conceptualization: the analysis of patterns and trends in science using networks of communications (by leaving out researchers themselves) and the formalization of the science system by citation links [12].

In this study, against the backdrop of the SSCT, we have outlined a new approach of creating global overlay maps using OpenAlex with raw and normalized overlay data. We have provided six different base maps that readers can use to overlay their own data such as national, institutional, or individual data. To exemplify the use of overlays with the base maps, we have provided six maps (data of the authors and four research institutes) using raw and normalized overlay data. The raw overlay maps in Fig 3 show that the overlay maps can be used to explore the topics of publication sets although one needs to be cautious when surprising concepts appear. The maps are suitable to explore the topics of different types of units of analysis, e.g., individuals and research institutes.

Our general impression is that the raw overlay maps emphasize higher level concepts more, and normalized overlay maps emphasize lower level concepts more. This can be explained as follows: Many more papers are assigned to higher level concepts. Thus, the higher level concepts are more visible on the raw overlay maps. Normalization of the overlay maps compares the focal unit's activity to the world's activity. Thereby, the higher level (more general) concepts lose prominence compared to lower level (more specialized) concepts. If one is interested in comparing different research units, normalized maps seem to be better suited than maps that are non-normalized. Both maps (normalized and non-normalized) provide different perspectives when gaining information about single research units. A more general perspective can be obtained using the non-normalized maps while a more detailed perspective is provided by the normalized maps.

The main advantage of our approach to consider OpenAlex data to produce overlay maps is that the underlying data can be used without any restrictions. Global base maps (covering the complete science system) and overlay data can be produced without any licensing costs. The underlying data are freely available. These advantages are accompanied by some limitations of the approach. One of the main limitations concerns the concepts provided by OpenAlex. The visual inspection of our individual maps revealed that some concepts are erroneously assigned to papers. We also found out that some concepts are only partly correct; especially the

additions in parentheses to labels were frequently erroneous or at least misleading. We hope that the reliability and validity of the process at OpenAlex to assign concepts to publications will be further improved in the near future. The announced new topics classification system (see: https://groups.google.com/g/openalex-users/c/2yE1jie_D3s/m/c3j9UYiLBgAJ?utm_medium= email&utm_source=footer) might provide improved assignments of documents in the future that will be available besides the concepts. As algorithmically assigned topics might not be much better than the concepts, base maps could also be constructed on the journal level [39]. However, this would exclude non-journal publications (e.g., books and conference proceedings).

Another limitation of our approach concerns the normalization method proposed here, which may not be perfect. The method compares all concepts with each other irrespective of scientific field. Overlay maps using normalization with respect to the scientific field of the concepts might produce different views. In the current normalization procedure, the number of papers in each concept is divided by the number of all papers with a concept on the same level of the unit using multiplicative counting. Alternatively, one could use the total number of papers of the unit that have a concept. Since the normalization method can present rarely occurring concepts very prominently because comparably few publications were assigned to such concepts, users may introduce thresholds for the number of papers per concept that prevent concepts with too few papers to be displayed. The third limitation is related to the base maps that have been produced by us for recent years. Although these base maps are based on large publication sets, some concepts are located further away from most concepts. For generating some of the base maps' images, it was necessary to zoom into the maps and to cut the outliers. This limitation, however, concerns only the maps based on papers from comparably short time periods. Our results show that at least a ten-year period should be used when a base map is constructed to avoid the zooming.

The visualization approach based on OpenAlex data that we introduced in this study can be used for any science unit. In the results section, we revealed the approach for researchers and institutions, but it is also possible to produce the maps for other units, e.g., research groups, journals, and countries. If the publication set for the overlays (concerning groups, journals, countries etc.) can be reliably collected in OpenAlex, it can be visualized applying the base maps. The overlay approach is especially useful for comparisons of science units: Since the specific publication sets of the units are positioned against the backdrop of the (same) whole science system, similarities and differences between the units can be immediately observed. We demonstrated this advantage of the overlay approach by comparing the maps for RH and LB: The maps clearly point out that both have been active in nearly the same area (bibliometrics/ scientometrics), but RH has additionally been active in other areas (chemistry and physics).

With the present study, we tried to develop a workflow by which the user is in a good position to produce meaningful overlay maps without any licensing costs or fees. Producing overlay maps with our provided base maps only needs standard PC/laptop hardware. We also assume that the approach introduced in this study could (should) be improved in future studies. We think, e.g., that it would be interesting to consider a broader set of overlay data. The current approach visualizes publication output (i.e., number of publications) on certain research topics. Node colors could easily be changed from field assignment to average publication years or proportion of papers assigned to some (or any) sustainable development goal (SDG) of the underlying documents. As of March 2024, OpenAlex also provides SDG assignments to the indexed works. The number of publications published in reputable journals and the normalized citation impact of publications are other standard indicators in research evaluation processes and could be well integrated in overlay maps, e.g., by node coloring. However, paper and journal based citation metrics using field-normalization are currently not available in OpenAlex.

## Acknowledgments

The bibliometric data used in this paper are from a custom database of OpenAlex hosted at the German Kompetenznetzwerk Bibliometrie (KB, Competence Network Bibliometrics, see https://bibliometrie.info/, funded by BMBF via grant 01PQ17001). This manuscript is based on another study submitted to STI 2024 [40].

## Author Contributions

**Conceptualization:** Robin Haunschild.

**Data curation:** Robin Haunschild.

**Formal analysis:** Robin Haunschild.

**Investigation:** Robin Haunschild, Lutz Bornmann.

**Methodology:** Robin Haunschild.

**Project administration:** Robin Haunschild.

**Resources:** Robin Haunschild.

**Software:** Robin Haunschild.

**Supervision:** Robin Haunschild.

**Validation:** Robin Haunschild, Lutz Bornmann.

**Visualization:** Robin Haunschild.

**Writing – original draft:** Robin Haunschild, Lutz Bornmann.

**Writing – review & editing:** Robin Haunschild, Lutz Bornmann.

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
