## [Decision Letter · Decision Letter 0]

5 Jun 2024

PONE-D-24-14267The use of OpenAlex to produce meaningful bibliometric global overlay maps of science on the individual, institutional, and national levelsPLOS ONE

Dear Dr. Haunschild,

Thank you for submitting your manuscript to PLOS ONE. After careful consideration, we feel that it has merit but does not fully meet PLOS ONE’s publication criteria as it currently stands. Therefore, we invite you to submit a revised version of the manuscript that addresses the points raised during the review process.

Please carefully check the Reviewers’ comments and improve the manuscript. 

We look forward to receiving your revised manuscript.

Kind regards,

Agnieszka Konys, Ph.D.

Academic Editor

PLOS ONE

“Both authors currently serve as academic editors for PLOS ONE.”

3. Please note that your Data Availability Statement is currently missing the DOI/accession number of each dataset OR a direct link to access each database. If your manuscript is accepted for publication, you will be asked to provide these details on a very short timeline. We therefore suggest that you provide this information now, though we will not hold up the peer review process if you are unable.

Reviewers' comments:

Reviewer's Responses to Questions

**Comments to the Author**

1. Is the manuscript technically sound, and do the data support the conclusions?

Reviewer #1: Yes

Reviewer #2: Partly

2. Has the statistical analysis been performed appropriately and rigorously? 

Reviewer #1: Yes

Reviewer #2: Yes

3. Have the authors made all data underlying the findings in their manuscript fully available?

Reviewer #1: Yes

Reviewer #2: Yes

4. Is the manuscript presented in an intelligible fashion and written in standard English?

Reviewer #1: Yes

Reviewer #2: Yes

5. Review Comments to the Author

Reviewer #1: Dear Authors,

I hope this message finds you well. I would like to congratulate you on your manuscript detailing the use of OpenAlex for producing bibliometric overlay maps. This topic is of significant importance as it addresses the increasingly crucial role of bibliometrics in understanding the dynamics and evolution of scientific research. By employing OpenAlex, a freely accessible database, your work contributes to the democratization of scientific analysis, allowing a broader range of researchers to engage with and utilize bibliometric data. This approach not only enhances transparency in research evaluations but also facilitates more informed decisions in science policy and funding.

Your manuscript describes a technically sound piece of scientific research with data that supports the conclusions, based on the detailed review of the experimental design, data analysis, and conclusions.

- Experimental Design and Methods: The manuscript details the use of OpenAlex for producing bibliometric overlay maps, utilizing data snapshots from different time periods.

- Data Presentation and Statistical Analysis: The manuscript employs normalization procedures to compare different datasets effectively. These methods are clearly described, allowing for the reproducible analysis of overlay maps, which is crucial for validating the scientific claims made.

- Interpretation of Results and Conclusions: The conclusions drawn from the analyses seem appropriate and are based on the data presented.

- Reproducibility and Rigor: The manuscript provides sufficient information on the data sources and analysis methods, which supports the reproducibility of the research.

The statistical analysis in the manuscript has been conducted with a clear methodology, focusing on normalization procedures for overlay data, which is essential for comparing different datasets on bibliometric maps. Here are the specifics of the analysis:

- Normalization Procedures: The manuscript outlines a method for normalizing the overlay data by comparing the proportion of documents per concept in the focus dataset to that in the global dataset.

- Rigorous Application of Statistical Methods: The approach includes calculating the total number of documents for both the world and the focal unit, and the proportions of documents assigned to specific concepts.

- Comparison of Overlay Maps: The manuscript also discusses the application of these normalized values to overlay maps, which helps in assessing the relative emphasis on different research topics between datasets.

- Potential Limitations and Transparency: The authors acknowledge potential limitations in their normalization method, such as the non-discrimination between scientific fields and hierarchical levels, which adds a layer of transparency and critical evaluation to their statistical approach.

The manuscript is presented in a clear and generally correct English language, conducive to comprehension within the academic community. The text is structured to convey complex methodologies and analyses understandably. However, there are a few minor issues and typographical errors that should be addressed:

- There are instances of punctuation inconsistencies, such as missing commas in compound sentences which could slightly hinder readability.

- The manuscript occasionally employs jargon and technical terms without definitions for a potentially broader audience. It would benefit from either a glossary or brief in-text definitions.

- It would enhance clarity to consistently use either the full term or its abbreviation after the first mention, rather than alternating between them.

- Correct minor typographical errors, such as "loose prominence" which should be "lose prominence".

In conclusion, the manuscript is well-written but would benefit from a careful proofreading session to correct minor typographical and grammatical errors to meet the high standards expected in published research.

Best regards.

Reviewer #2: The base maps provide insights into various themes, topics, and global research areas or specific research units. The data for the paper was sourced from OpenAlex, and the maps were created by importing the data into VOSviewer. The authors processed the OpenAlex data to make it VOSviewer compliant. In the results section, the authors focused on the subject classification attributes presented in the six global base maps. However, it would be more relevant to discuss themes and topics, considering the existence of many well-established classification systems for subject areas. The authors should improve this perspective to enhance the potential applications of this study.

Furthermore, the authors mentioned that no cost or expense is required. However, the paper should specify the hardware facilities needed, such as CPU and RAM, as well as the processing time required. This information should have been explained in the thesis.

6. PLOS authors have the option to publish the peer review history of their article (what does this mean?). If published, this will include your full peer review and any attached files.

Reviewer #1: No

Reviewer #2: No

---

## [Author Response · Author response to Decision Letter 0]

8 Jul 2024

E-P1) Thank you for stating the following in the Competing Interests section:

“Both authors currently serve as academic editors for PLOS ONE.”

Ad E-P1) Thank you. We have updated the cover letter.

E-P2) Please note that your Data Availability Statement is currently missing the DOI/accession number of each dataset OR a direct link to access each database. If your manuscript is accepted for publication, you will be asked to provide these details on a very short timeline. We therefore suggest that you provide this information now, though we will not hold up the peer review process if you are unable.

Ad E-P2) We have used data from OpenAlex that are openly available from https://openalex.org/. Our base maps are available at: https://doi.org/10.17617/1.daf7-fq06.

E-P3) Please review your reference list to ensure that it is complete and correct. If you have cited papers that have been retracted, please include the rationale for doing so in the manuscript text, or remove these references and replace them with relevant current references. Any changes to the reference list should be mentioned in the rebuttal letter that accompanies your revised manuscript. If you need to cite a retracted article, indicate the article’s retracted status in the References list and also include a citation and full reference for the retraction notice.

Ad E-P3) We have changed the EndNote reference style. We are not aware of any retracted papers in our list of references.

Reviewer's Responses to Questions

1. Is the manuscript technically sound, and do the data support the conclusions?

Reviewer #1: Yes

Reviewer #2: Partly

Authors: Thank you.

2. Has the statistical analysis been performed appropriately and rigorously?

Reviewer #1: Yes

Reviewer #2: Yes

Authors: Thank you.

3. Have the authors made all data underlying the findings in their manuscript fully available?

Reviewer #1: Yes

Reviewer #2: Yes

Authors: Thank you.

4. Is the manuscript presented in an intelligible fashion and written in standard English?

Reviewer #1: Yes

Reviewer #2: Yes

Authors: Thank you.

5. Review Comments to the Author

Reviewer #1: 

R1-P1) Dear Authors, I hope this message finds you well. I would like to congratulate you on your manuscript detailing the use of OpenAlex for producing bibliometric overlay maps. This topic is of significant importance as it addresses the increasingly crucial role of bibliometrics in understanding the dynamics and evolution of scientific research. By employing OpenAlex, a freely accessible database, your work contributes to the democratization of scientific analysis, allowing a broader range of researchers to engage with and utilize bibliometric data. This approach not only enhances transparency in research evaluations but also facilitates more informed decisions in science policy and funding.

Your manuscript describes a technically sound piece of scientific research with data that supports the conclusions, based on the detailed review of the experimental design, data analysis, and conclusions.

Experimental Design and Methods: The manuscript details the use of OpenAlex for producing bibliometric overlay maps, utilizing data snapshots from different time periods.

Data Presentation and Statistical Analysis: The manuscript employs normalization procedures to compare different datasets effectively. These methods are clearly described, allowing for the reproducible analysis of overlay maps, which is crucial for validating the scientific claims made.

Interpretation of Results and Conclusions: The conclusions drawn from the analyses seem appropriate and are based on the data presented.

Reproducibility and Rigor: The manuscript provides sufficient information on the data sources and analysis methods, which supports the reproducibility of the research.

The statistical analysis in the manuscript has been conducted with a clear methodology, focusing on normalization procedures for overlay data, which is essential for comparing different datasets on bibliometric maps. Here are the specifics of the analysis:

Normalization Procedures: The manuscript outlines a method for normalizing the overlay data by comparing the proportion of documents per concept in the focus dataset to that in the global dataset.

Rigorous Application of Statistical Methods: The approach includes calculating the total number of documents for both the world and the focal unit, and the proportions of documents assigned to specific concepts.

Comparison of Overlay Maps: The manuscript also discusses the application of these normalized values to overlay maps, which helps in assessing the relative emphasis on different research topics between datasets.

Potential Limitations and Transparency: The authors acknowledge potential limitations in their normalization method, such as the non-discrimination between scientific fields and hierarchical levels, which adds a layer of transparency and critical evaluation to their statistical approach.

Ad R1-P1) Thank you. We appreciate your positive comments very much.

R1-P2) The manuscript is presented in a clear and generally correct English language, conducive to comprehension within the academic community. The text is structured to convey complex methodologies and analyses understandably. However, there are a few minor issues and typographical errors that should be addressed:

- There are instances of punctuation inconsistencies, such as missing commas in compound sentences which could slightly hinder readability.

- The manuscript occasionally employs jargon and technical terms without definitions for a potentially broader audience. It would benefit from either a glossary or brief in-text definitions.

- It would enhance clarity to consistently use either the full term or its abbreviation after the first mention, rather than alternating between them.

- Correct minor typographical errors, such as "loose prominence" which should be "lose prominence".

In conclusion, the manuscript is well-written but would benefit from a careful proofreading session to correct minor typographical and grammatical errors to meet the high standards expected in published research.

Ad R1-P2) We have corrected the typographical and grammatical errors and punctuation inconsistencies that we have spotted. We are not sure which jargon the reviewer refers to. The concepts have a Wikidata entry with an explanation. 

Reviewer #2: 

R2-P1) The base maps provide insights into various themes, topics, and global research areas or specific research units. The data for the paper was sourced from OpenAlex, and the maps were created by importing the data into VOSviewer. The authors processed the OpenAlex data to make it VOSviewer compliant. In the results section, the authors focused on the subject classification attributes presented in the six global base maps. However, it would be more relevant to discuss themes and topics, considering the existence of many well-established classification systems for subject areas. The authors should improve this perspective to enhance the potential applications of this study.

Ad R2-P1) Thank you. We have added a description and discussion of the other subject classifications now available in OpenAlex. We think that external classification systems are not helpful for the focus of our manuscript as readers should also be able to easily reproduce our methodology. The use of closed access subject classifications would be counterproductive to our aim to provide openly reusable global overly maps of science.

R2-P2) Furthermore, the authors mentioned that no cost or expense is required. However, the paper should specify the hardware facilities needed, such as CPU and RAM, as well as the processing time required. This information should have been explained in the thesis.

Ad R2-P2) Thank you, we have specified that we mean licensing costs. We also added information about the employed computational resources and timing data.

---

## [Editor Report · Decision Letter 1]

17 Jul 2024

The use of OpenAlex to produce meaningful bibliometric global overlay maps of science on the individual, institutional, and national levels

PONE-D-24-14267R1

Dear Dr. Haunschild,

We’re pleased to inform you that your manuscript has been judged scientifically suitable for publication and will be formally accepted for publication once it meets all outstanding technical requirements.

Kind regards,

Agnieszka Konys, Ph.D.

Academic Editor

PLOS ONE
---

## [Editor Report · Acceptance letter]

13 Aug 2024

PONE-D-24-14267R1 

PLOS ONE

Dear Dr. Haunschild, 

I'm pleased to inform you that your manuscript has been deemed suitable for publication in PLOS ONE. Congratulations! Your manuscript is now being handed over to our production team.

Kind regards, 

on behalf of

Dr. Agnieszka Konys 

Academic Editor

PLOS ONE